# Analysis of LLL System Properties for Different Excitation Parameters

**Krzysztof Wandachowicz** *,† , **Małgorzata Zalesińska** † and **Przemysław Otomański**

Institute of Electrical Engineering and Electronics, Faculty of Control, Robotics and Electrical Engineering, Poznan University of Technology, Piotrowo Street, No. 3a, 60-965 Poznan, Poland; Malgorzata.Zalesinska@put.poznan.pl (M.Z.); Przemyslaw.Otomanski@put.poznan.pl (P.O.)
* Correspondence: Krzysztof.Wandachowicz@put.poznan.pl
† These authors contributed equally to this work.

**Abstract:** Photoluminescent strips forming a Low Location Lighting (LLL) system are the primary method for marking escape routes on passenger ships. The LLL system can be built as a self-luminous system (powered by electricity) or made as a series of strips made of photoluminescent materials, which glow and indicate the escape route after the loss of basic and emergency lighting. To ensure correct visual guidance, these strips must be installed at specific locations in the passageways and achieve appropriate photometric parameters after a certain time from their activation. The properties of the LLL system depend on the type of luminescent material used, the excitation source, and the exposure parameters. This paper presents the results of laboratory tests on two types of photoluminescent materials used for the construction of LLL systems. We recorded the change in luminance after the loss of excitation and measured the luminance values obtained 10 and 60 min after the loss of excitation under exposure to light sources commonly used for interior lighting on passenger ships. It turns out that replacing fluorescent lamps with LED lamps can reduce the luminance of the LLL system.

**Keywords:** lighting engineering; photoluminescent lighting system; Low Location Lighting system

## 1. Introduction

The International Convention for the Safety of Life at Sea 1974 SOLAS [1] sets safety regulations for all vessels. According to the provisions of the Convention, all passenger ships designed to carry more than 36 passengers must have a Low Location Lighting (LLL) system. This is the last safety system that should allow for finding the escape route even in heavy smoke. For this reason, the electric lighting system and photoluminescent strips are located at a low height above the ground. The LLL system based on photoluminescent strips is widely used on passenger ships and ferries due to its reliability (electrical power not needed for operation).

The method for marking escape routes on passenger ships, the required luminance values after 10 and 60 min after the lights go off and the minimum number of measurement points are specified in ISO 15 370 [2].

According to the requirements presented in [2], the strips should be installed on corridor walls, walls, or steps of staircases and doors leading to stairs, to exits from decks, and separating main corridor zones. In corridors, except for cabin doors, recesses less than 2 m wide, and connections to other passageways, the photoluminescent strips shall be so placed as to provide continuous visual information on the escape route after the loss of basic lighting. The maximum discontinuity of the strips shall not exceed 2 m. Photoluminescent strips should be installed 300 mm above the floor or at floor level 150 mm from the wall. In stairways leading to emergency exits, the LLL system should be installed on the walls at no more than 300 mm above the floor at each step. In any case, the LLL system should be installed at least on one side of the passageway if it is less than 2 m wide and on both sides if

it is more than 2 m wide. The marking of doors on escape routes should indicate the location of the handle or grip for opening it. Figure 1 shows the location of photoluminescent strips as required by the standard [2] on an example corridor and staircase.

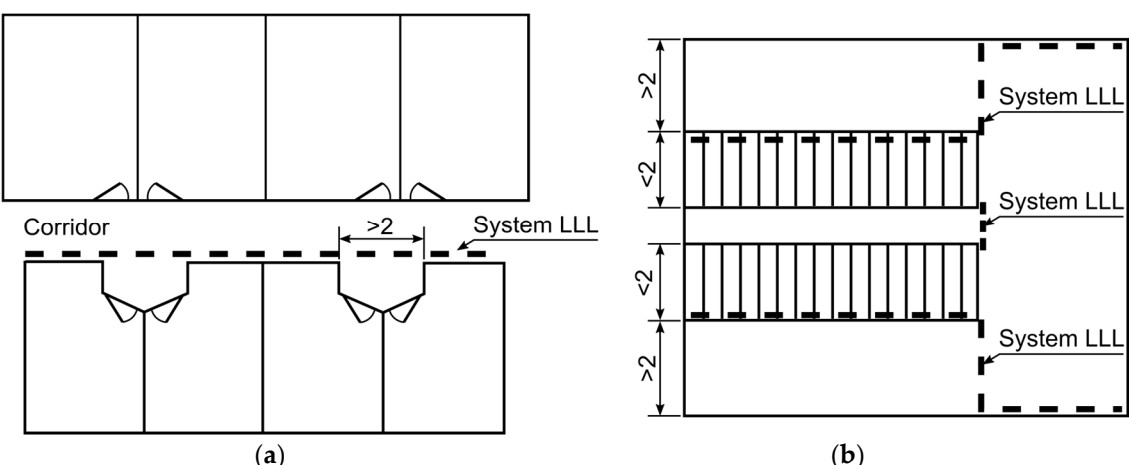

|            |            |
|:----------:|:----------:|
| (**a**)    | (**b**)    |

**Figure 1.** Photoluminescent stripe location as required by ISO 15370:2010 [2]: (**a**) Corridor with recesses wider than 2 m; (**b**) Stairway wider than 2 m.

The LLL system must ensure visual guidance for all the time necessary for disembarkation from all decks of the ship. The LLL system should be operational at least 60 min after activation. The photoluminescent materials should keep a luminance (L) of not less than 15 mcd/m$^2$ at 10 min and 2 mcd/m$^2$ at 60 min after the basic lighting goes off.

Under the ISO 15,370 and IMO [2,3] requirements, the LLL system should be tested at least every five years for correct functioning. Measurements verifying the photometric properties of the LLL system in real conditions are time-consuming measurements and carried out during normal ship operation. Figure 2 presents the stands for measuring the luminance of the LLL system in real-life operating conditions.

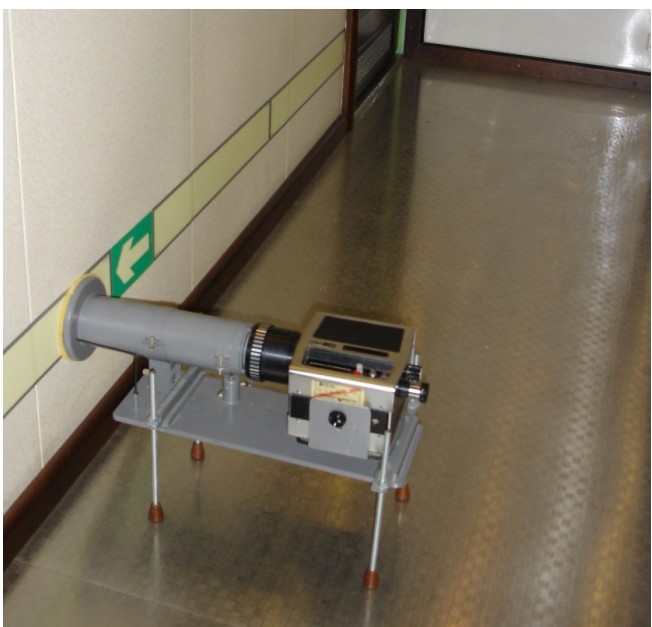

**Figure 2.** LLL system luminance measurement in real-life operating conditions.

The measurement result, and thus the photometric properties of the photoluminescent strips, are influenced by such factors as the type of material, the spectral distribution of the light source illuminating the strips, the value of the illuminance on the strip surface, and

the exposure time. If the influence of individual factors is known, a preliminary assessment of the properties of applied photoluminescent strips is possible together with determining the location of measurement points.

ISO 17398:2004 and ISO 15370:2010 are currently used to assess a material's photometric properties. However, in neither of these standards do the measurement procedures allow the assessment of the properties of photoluminescent materials under real-life conditions. The classification of materials presented in the standard [4] only allows a comparison of the properties of materials tested in the laboratory. In this case a xenon lamp of max. 500 W illuminating the tested sample for 5 min. and generating illuminance on the surface of the sample equal to 200 lx is the excitation source. The standard [2] allows for a more comprehensive assessment of the properties of photoluminescent materials by determining for a given material, in laboratory conditions, the minimum level of illuminance which allows providing adequate luminance values during 10 and 60 min after the excitation source goes off. According to the requirements of the standard [2], the excitation source is an 8 W fluorescent lamp being an F 2 illuminant with a colour temperature of 4100 K that provides an illuminance of 25 lx on the strip surface.

On passenger ships, photoluminescent strips are used in extremely varied lighting conditions [5–7], thus it is necessary to run laboratory tests using the most commonly used light sources. The following values were determined during the tests:

- Minimum illuminance on the strip surface which satisfies the normative requirements [2];
- Luminance changes over time after the excitation source goes off;
- Luminance values measured 10 and 60 min after the excitation source goes off.

The obtained results allow for a thorough assessment of the photometric properties of materials, which may lead to significant improvement of measurements in the field conditions (on ships).

## 2. Laboratory Tests of LLL System

Photoluminescent strips were tested in a photometric laboratory according to normative requirements. For measurement purposes, we created a laboratory stand with a luminance meter and a computer for recording data. Measurements were performed for two photoluminescent materials. The tested samples came from two passenger ferries operating on routes from Poland to Scandinavia. There are photoluminescent materials on the market with different properties. Cheaper materials have average photometric properties and low luminance values after the disappearance of the excitation source. Zinc sulphide is the main, active component of these types of materials. More expensive materials have improved photometric properties. Its active components include rare earth elements. The technologies used in the production of this type of material allow for achieving high luminance levels and long luminance decay times. Table 1 presents a description of the two types of materials that were used for the tests.

**Table 1.** Description of the materials used for the tests.

| Material Name | Description |
|---|---|
| Material no. 1 | Cheap; Average photometric properties and low luminance values; The main, active component is Zinc sulphide. |
| Material no. 2 | More expensive; Improved photometric properties; Its active components include rare earth elements. |

The investigated photoluminescent materials were excited with light sources of different spectral distributions at different illuminance values. The tests were performed with the following lamps (where: CCT—Correlated Color Temperature, CRI—Colour Rendering Index):

- H—Halogen incandescent lamp;

- FL640—Fluorescent lamp with colour code 640—CCT 4000 K, CRI 60;
- FL840—Fluorescent lamp with colour code 840—CCT 4000 K, CRI 80;
- FL830—Fluorescent lamp with colour code 830—CCT 3000 K, CRI 80;
- LED840—LED lamp with colour code 840—CCT 4000 K, CRI 80;
- LED830—LED lamp with colour code 830—CCT 3000 K, CRI 80;

Figure 3 presents the spectral distributions of the light sources.

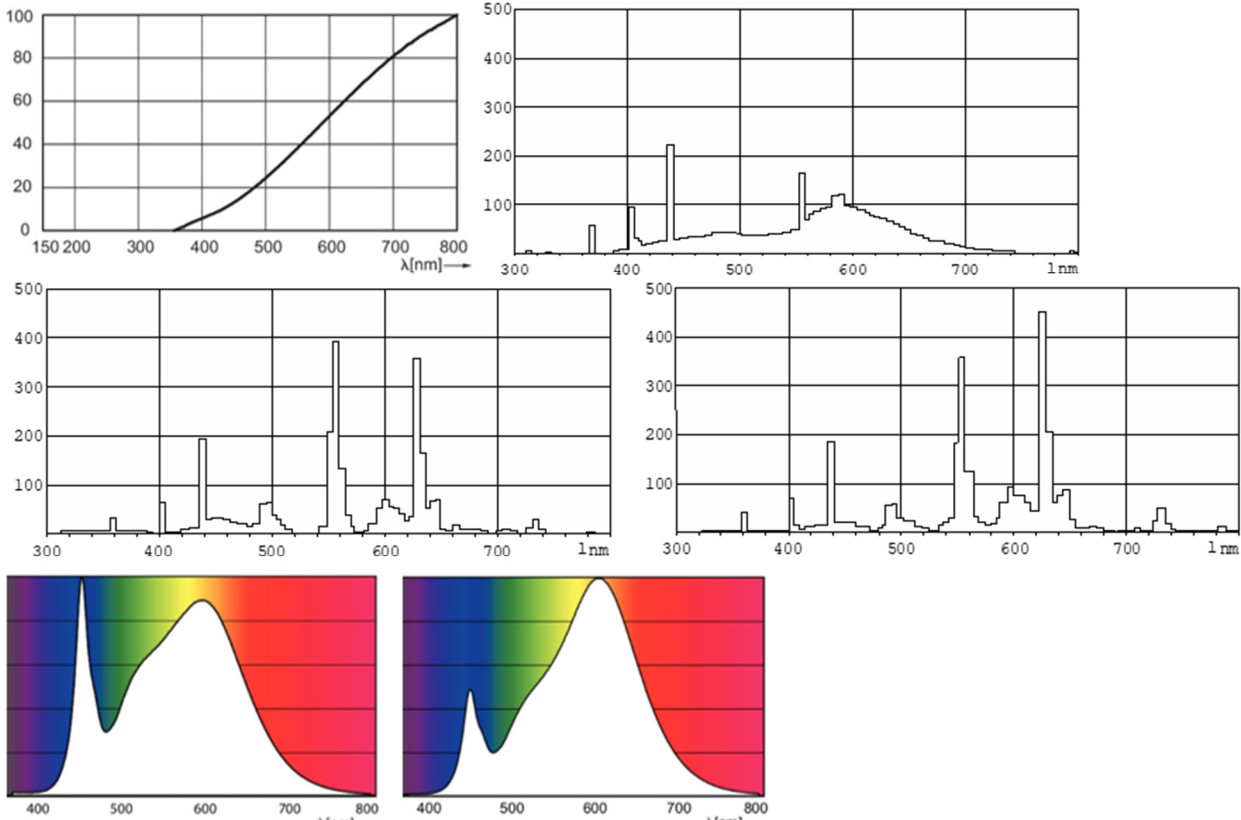

**Figure 3.** The spectral distributions of the light sources [8].

Under the requirements of the standard [2], all the measurements were performed 24 h after the samples went into darkness. Based on previous measurement experience [5,6], for a given excitation source, appropriate illuminance levels on the sample surface were determined each time. In each case, we determined luminance decay curves after switching the excitation source off and read characteristic luminance values.

Figure 4 presents the results of luminance measurements of the investigated photoluminescent materials under different lighting conditions. Figure 5 presents example luminance decay curves for the fluorescent lamp with colour 840, for different values of illuminance on the surfaces of the tested materials. Figure 6 presents luminance decay curves for all tested lamps for two values of of illuminance on the surfaces of the tested materials, 12.5 lx and 25 lx.

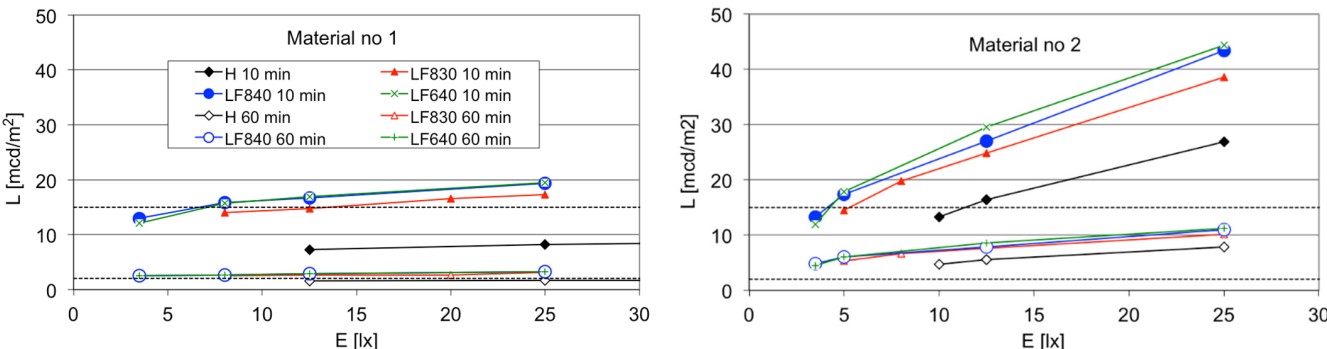

**Figure 4.** Changes in luminance values measured 10 and 60 min after the excitation source goes off for four lamps and two materials.

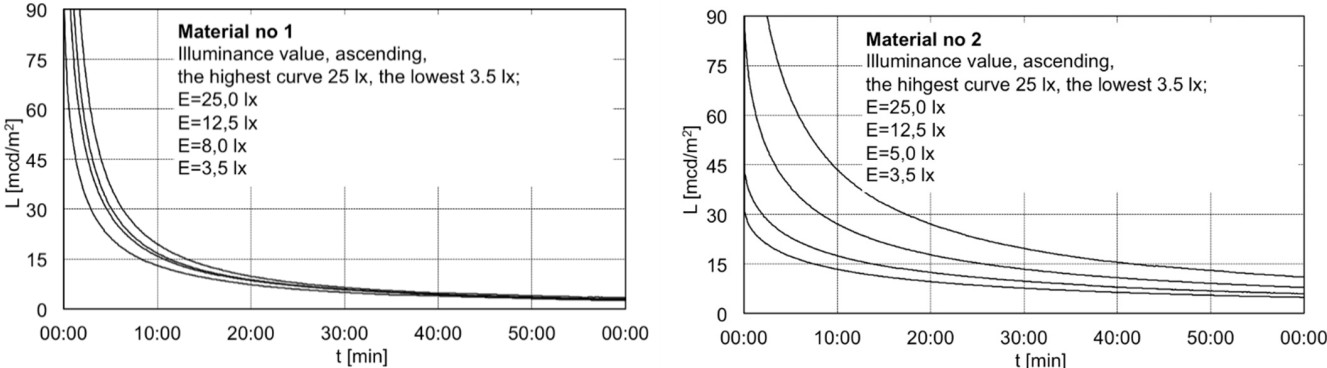

**Figure 5.** Example curves presenting luminance change in time after the disappearance of the excitation source for the 840 fluorescent lamp.

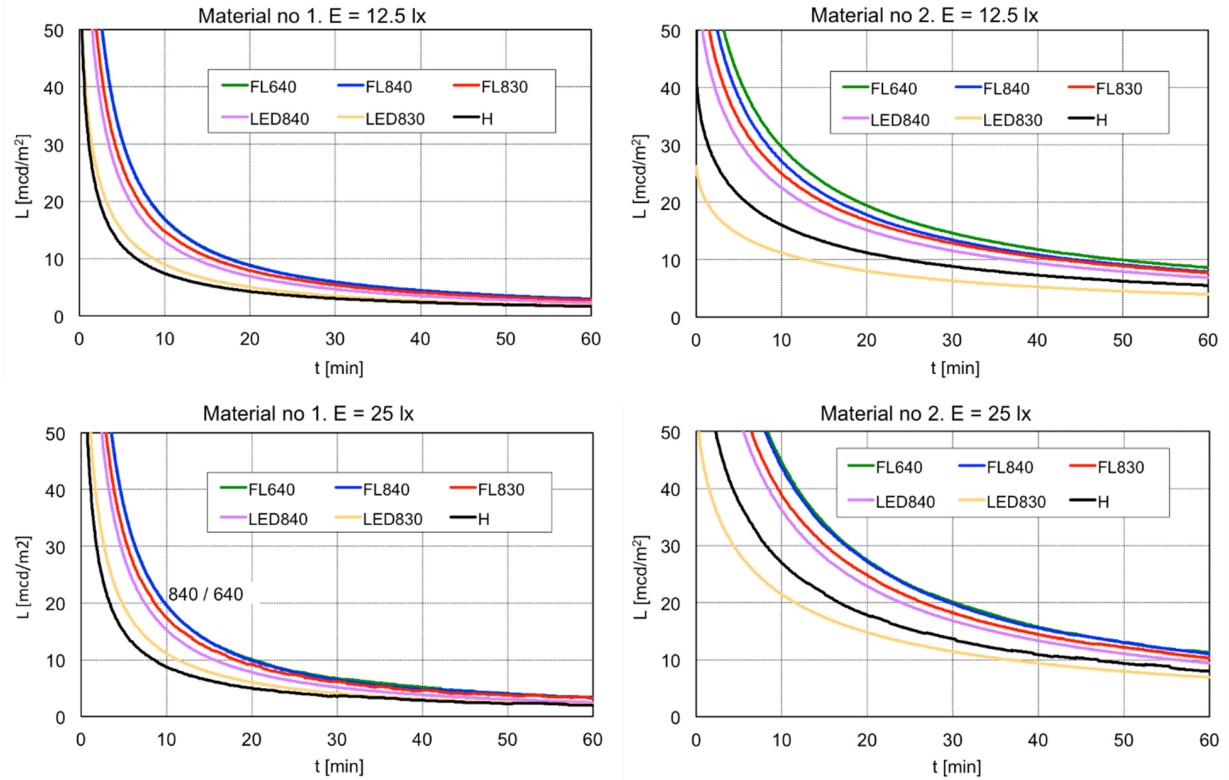

**Figure 6.** Luminance change in time after the disappearance of the excitation source for all tested lamps and two values of of illuminance on the surfaces of the tested materials 12.5 lx and 25 lx.

The conducted laboratory measurements showed that the photometric properties of materials are influenced by the spectral distribution of the excitation source, the value of illuminance at the tested sample surface, and the chemical composition of the material. For material no. 2, which has improved photometric properties, the influence of illuminance was greater than for material no. 1 (Figure 4). The differences between the measured luminance values in the tested samples increased with the level of illumination of the material surface. Table 2 shows the luminance values measured 10 and 60 min after the disappearance of the excitation source.

**Table 2.** The luminance values measured 10 and 60 min after the disappearance of the excitation source (values that do not meet the requirements [2] have been marked by background grey colour).

| L [mcd/m$^2$] | | Material No. 1 | | Material No. 2 | |
|---|---|---|---|---|---|
| | | E = 12.5 lx | E = 25 lx | E = 12.5 lx | E = 25 lx |
| Halogen | 10 min | 7.3 | 8.7 | 16 | 26.9 |
| | 60 min | 1.6 | 2 | 5.5 | 7.9 |
| LF 830 | 10 min | 14.7 | 17.3 | 24.9 | 38.6 |
| | 60 min | 2.7 | 3.2 | 7.6 | 10.2 |
| LF 840 | 10 min | 16.7 | 19.4 | 27 | 43.4 |
| | 60 min | 2.9 | 3.3 | 7.8 | 11 |
| LF 640 | 10 min | 16.9 | 19.5 | 29.5 | 44.4 |
| | 60 min | 2.9 | 3.3 | 8.6 | 11.2 |
| LED 830 | 10 min | 8.9 | 11.1 | 11.2 | 21.4 |
| | 60 min | 1.8 | 2.1 | 4 | 6.9 |
| LED 840 | 10 min | 13 | 15.3 | 22.5 | 36.2 |
| | 60 min | 2.3 | 2.5 | 6.9 | 9.4 |

For a fluorescent lamp with an illumination level of 25 lux, after 10 min from the disappearance of excitation luminance of material no. 2 was approximately 2.2 times higher than that of material no. 1, and after 60 min it was approximately 3.2 times higher. For the LED lamps, it was 1.9 ÷ 2.4 and 3.3 ÷ 3.8 respectively. For the halogen bulb, it was 3.2 and 4.6 respectively. At an illumination level of 12.5 lux, the material luminance after 10 min was about 1.7 times higher for fluorescent lamps, 1.2 ÷ 1.7 times higher for LED lamps and 2.2 for the incandescent bulb. After 60 min, it was 2.8, 2.2 ÷ 3 and 3.5, respectively. For material no. 1, the illuminance level had less influence on the shape of the luminance change curve as a function of time after the disappearance of the excitation source (Figure 5). Material no. 1 was also characterised by a faster decrease in luminance after the excitation source went off. Based on the measurement results, for the sources of excitation under consideration, we determined the minimum values of illuminance on the surfaces of the tested materials allowing them to fulfil the requirements of the standard [2]. The determined values are presented in Table 3.

**Table 3.** The minimum values of illuminance on the surfaces of the tested materials allowing them to fulfil the requirements of the standard [2].

| Lamp | Material No. 1 $E_{min}$ [lx] | Material No. 2 $E_{min}$ [lx] |
|---|---|---|
| Halogen lamp | 4915 | 11 |
| LF 830 | 14 | 5 |
| LF 840 | 10 | 4 |
| LF 640 | 10 | 4 |
| LED 830 | 32 | 18 |
| LED 840 | 24 | 8 |

## 3. Discussion

Analysis of change in the luminance of photoluminescent strips showed the influence of the illumination intensity and spectral distribution of light sources on the luminance values obtained 10 and 60 min after the excitation source was switched off (Table 2).

The standard requirements [2] are not met for material no. 1 with an incandescent light source and LED830 lamp regardless of the illuminance value on the surface of the material. Tests of LLL strips under laboratory conditions are performed according to the requirements of the standard [2] at an illumination level of 25 lx at the material surface obtained from a fluorescent lamp with a colour temperature of 3000 K. However, an illumination level of 25 lx is insufficient to obtain the required luminance values when excited with incandescent light sources and LED830 lamps. Especially, after 10 min of excitation decay, the luminance values obtained do not meet the requirements of the standard [2]. When illuminated by incandescent lamps and LED830 lamps, the illuminance values on the surfaces of standard LLL strips (with zinc sulphide as the active component) should be higher than the illuminance values for performing laboratory tests. With the illuminance level increased above 25 lux, the required luminance values might be achieved. Incandescent light sources (such as halogen bulbs) are still used on some passenger ships.

For a fluorescent light source, the requirements of the standard [2] were met for each level of illumination only for material no. 2 (with rare earth elements as the active component). For material no. 1 the requirements of the standard [1] are not met for a fluorescent lamp in colour 830, the illumination level of 12.5 lx, and after 10 min from the decay of excitation. Passenger ship decks are treated like hotel rooms and warm light colour is preferred (colour temperature 2700–300 K). As shown, a warm colour does not always allow sufficiently strong illumination of the photoluminescent strips. Higher illumination levels are necessary or, in evacuation zones, light sources with cooler colours should be used.

The results of the measurements indicate, surprisingly, that the LED lamps are less effective in stimulating the photoluminescent material (Tables 2 and 3). The requirements of the standard [2] were not met for LED830 lamp. For LED840 lamp the requirements were met only for material no. 2.

The measurements showed a significant influence of the spectral distribution of light sources on the luminance of photoluminescent materials and on the shape of the luminance decrease curve as a function of time from turning the lighting off. Light sources which radiate more energy in the short-wave part of the spectrum (fluorescent lamps and LED lamps with a colour temperature of 4000 K) allow for obtaining higher luminance values regardless of the level of illuminance on the material surface.

For light sources with low colour temperature values, the influence of the illuminance level on the achieved strip luminance value strips after excitation decay is more significant than for lamps with higher colour temperature values. With the illumination level doubled for a halogen bulb, an increase in luminance values of 19% (material number 1) and 68% (material number 2) after 10 min and 25% (material number 1) and 44% (material number 2) after 60 min is obtained. For a dual-band, luminophore fluorescent lamp with a colour temperature of 4000 K, a relatively smaller increase of luminance values was obtained for the same difference in illuminance levels and it was 15% (material number 1) and 51% (material number 2) and 14% (material number 1) and 30% (material number 2) respectively.

The use of a material with better photometric properties makes it possible to obtain higher luminance values at each illuminance level and thus meet the normative requirements under less favourable lighting conditions, e.g., lower illuminance levels for the LLL system.

## 4. Conclusions

Laboratory tests make it possible to assess the properties of photoluminescent materials. We demonstrated a significant influence of illumination level, exposure time, spectral distribution and chemical composition of photoluminescent materials on photometric

properties of LLL strips. The use of materials with average photometric properties and low luminance values after excitation decay in many cases leads to the LLL system failing to meet the requirements of ISO 15370 under the given illumination conditions. The measurement results and conclusions from our tests facilitate the measurement procedures for a LLL system operated in real lighting conditions and allow for the formulation of guidelines to improve their operation. The description of material properties presented here does not cover the changes that occur during operation.

Our team has many years of experience in measuring LLL systems on passenger ships. The illuminance values we measured on the surface of the photoluminescent stripes ranged from about 10 to about 50 lux. We have measured values less than 12.5 Lux and 25 Lux many times. In such cases, the LLL system usually did not meet the requirements [2] (Table 3).

Currently, fluorescent lamps are being replaced with LED lamps. The use of LED lamps increases the energy efficiency of lighting installations. It turns out, however, that when changing to LED lighting, care should be taken to use higher levels of illuminance (Table 3). Our experience shows that people responsible for maintaining the LLL system in a proper condition do not have sufficient knowledge on this subject. Neglects in this regard may lead to deterioration of safety on passenger ships.

**Author Contributions:** Conceptualization, M.Z. and K.W.; methodology, M.Z. and K.W.; validation, M.Z. and K.W.; investigation, M.Z. and K.W.; data curation, K.W. and P.O.; writing—original draft preparation, K.W.; writing—review and editing, K.W.; visualization, K.W.; supervision, K.W.; funding acquisition, K.W. All authors have read and agreed to the published version of the manuscript.

**Funding:** This research was funded by the Polish Ministry of Education and Science from the Research and Development Subsidy no 0212/SBAD/0539.

**Institutional Review Board Statement:** Not applicable.

**Informed Consent Statement:** Not applicable.

**Data Availability Statement:** https://www.dropbox.com/sh/2yhlvm0krjheivx/AACi1hFghY-NXFFY8 wmeJstfa?dl=0.

**Conflicts of Interest:** The authors declare no conflict of interest.

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
