# Peer review of "Analysis of LLL System Properties for Different Excitation Parameters"

_energies, doi:10.3390/en14227723_

Round 1

Reviewer 1 Report

  • Similar published paper in Polish by 2 out of the 3 authors:

Wandachowicz K., Zalesińska M.: Badanie własności pasów fotoluminescencyjnych. Przegląd Elektrotechniczny, ISSN 263 1731-6106, R.5 NR 1/2007 pp.59-62.

  • Define LLL the first time it appears in the abstract.
  • Define more accurately the difference in the composition/ quality of the 2 photoluminescent materials used for the testing.

Author Response

Point 1. Similar published paper in Polish by 2 out of the 3 authors: Wandachowicz K., Zalesińska M.: Badanie własności pasów fotoluminescencyjnych. Przegląd Elektrotechniczny, ISSN 263 1731-6106, R.5 NR 1/2007 pp.59-62.

Response 1. The paper from 2007 contains studies done for halogen bulbs and fluorescent lamps. In the new paper, we present the research done for LEDs and compare them with the results obtained for fluorescent lamps. In addition, we present the minimum values of illuminance on the surfaces of the tested materials allowing them to fulfil the requirements of the standard.

Point 2. Define LLL the first time it appears in the abstract.

Response 2. We have added information to the abstract.

Point 3. Define more accurately the difference in the composition/ quality of the 2 photoluminescent materials used for the testing.

Response 3. We changed the text of the paper, to which we added information about the materials and a table with their description. We do not want to name the manufacturer of the materials.

Reviewer 2 Report

I have read your literature, and you mainly made experiments on the luminous characteristics of the light-emitting strips. This is a very important job for people's life safety! This is a very meaningful thing! But after reading, I am interested in the following points, hoping to answer:
1) Can you briefly describe your experimental device and how it conducts the experiment? Whether there are other coverings on the surface of the light-emitting strip you are measuring, such as plastic film. In other words, do you need plastic films to protect fluorescent substances in actual safety facilities? How can you prove that your experimental data is accurate and that it is under safe conditions?
2) Can you tell us about your experimental plan (Figure 2) and it will definitely meet the real-life operating conditions?

Reviewer 3 Report

I have analysed this manuscript which deals with safety lighting systems.

This paper is interesting in trying to establish a uniform set of guidelines for LLL systems.

The hypothesis and methodology are well established and the graphs and presentations are quite sufficient.

I see no important flaw in this article therefore I recommend its publication

The antecedents of the manuscript are sufficient. The examples selected are adequate.

The author convincingly demonstrates the scope and possibilities of this research. The contribution is interesting for all safety regulations and specially at Sea.

The references are accurate and correct.

Summary of evaluation. Favourable because of the expertise and knowledge demonstrated. Acceptance.
